# Genome-Wide Association Study (GWAS) for Left Displaced Abomasum in Highly Productive Russian Holstein Cattle

**DOI:** 10.3390/ani14192795

**Published:** 2024-09-27

**Authors:** Kirill Plemyashov, Anna Krutikova, Angelina Belikova, Tatiana Kuznetsova, Boris Semenov

**Affiliations:** Faculty of Veterinary Medicine, St. Petersburg State University of Veterinary Medicine, 196084 St. Petersburg, Russia

**Keywords:** LDA, GWAS, ROH, Holstein cattle

## Abstract

**Simple Summary:**

Left displaced abomasum is a common disease among Holstein cattle that is causing significant economic losses. The exact causes of the disease are unknown, but it is assumed that LDA is multifactorial in nature. Predisposition to the displacement has been shown to be inherited. We conducted a genome-wide study to identify statistical associations between certain mutations and left displaced abomasum in Holstein cows from several farms in the Leningrad region. We found suggestive statistical associations with this disease and several mutations located in a set of genes involved in a variety of physiological processes. Some of these processes may be directly related to the development of the disease being studied. In addition, we conducted runs of homozygosity analysis and found that cows with displaced abomasum are more likely than healthy cows to carry a specific state of genomic regions containing genes associated with body shape, including width. This may indicate the role of this trait in the development of the disease.

**Abstract:**

Left displaced abomasum (LDA) is a multifactorial disease of cattle that occurs mainly during the transition postpartum period and is characterized by a decrease in milk production and an increased risk of culling. Several studies have been conducted confirming the hereditary nature of predisposition to this disease. The aim of our study is to identify genetic associations characterizing the genomic variability of susceptibility to LDA in Holstein cattle of the Leningrad region of the Russian Federation. The objects of this study were 360 highly productive dairy cows divided into two groups: animals with LDA, and healthy ones (control). Runs of homozygosity analysis revealed one ROH on BTA13 that was found to be significantly more prevalent in the group of animals with LDA than in the healthy group. Fourteen candidate SNPs were found to be nominally associated with left displacement of the abomasum (*p*-value < 1 × 10^−4^). When performing functional annotation of genes containing associated polymorphisms or located close to them, candidate genes presumably associated with the development of LDA were identified: *ABCB11*, *SRP72*, *RGS18*, *SOX4*, *GSG1L*, *FBXL19*, and *PNPLA4.*

## 1. Introduction

Left displaced abomasum (LDA) is a multifactorial disease of cattle that occurs mainly during the transition postpartum period [1,2]. In the left form of displacement, the abomasum occupies a caudodorsal position, located between the rumen and the left abdominal wall. The pathology is characterized by a decrease in milk production and an increased risk of culling. The most effective way to treat this pathology is surgery, which significantly increases the economic costs of milk production. The median incidence of left displaced abomasum among Holstein cattle based on 12 studies is 2.71% [3], which makes this pathology an important threat to the economic stability of dairy livestock farming.

Numerous studies have established that left displacement of the abomasum occurs more often (75% of cases) than right displaced abomasum (RDA) [4,5]. Right displacement is more likely to be persistent, in contrast to the left-sided form, which significantly increases harm caused by the pathology. It has also been established that the right form is more acute than the left one, and requires prompt surgical treatment. However, due to its frequent occurrence, the economic damage from LDA significantly exceeds the losses from RDA.

Despite the widespread occurrence of LDA, its etiology remains not fully understood. There are several hypotheses explaining LDA etiology. It is believed that one of the most important prerequisites for displacement of the abomasum is atony arising as a result of inadequate feeding that does not correspond to the physiological status, leading to the accumulation of gas and liquid in the organ cavity, since this symptom occurs in more than 80% of all cases [4,6,7]. According to the hypothesis, gas appears in the abomasum body and, with an increase in its volume, shifts abomasum cranially. The flow of gas toward the pyloric region causes displacement of the abomasum to the cranial dorsal side and then its movement to the left side of the abdominal wall [8]. Occurrence of left displaced abomasum was also associated with the elliptical shape of the abdominal cavity section [9] and negative energy balance [10]. It has also been demonstrated that an increase in beta-hydroxybutyrate (BHB) and aspartate aminotransferase levels in the first two weeks after calving is a predictor for the occurrence of the described pathology [11,12,13], as well as low concentrations of liver-secreted IGF1 together with increased non-esterified fatty acids (NEFA) and BHB [13]. Cows with left displaced abomasum often have comorbidities such as ketosis, mastitis, metritis, endometritis, and fatty liver [4,14].

Heritability for LDA has been estimated in Holstein cattle to be between 0.1 and 0.31 [2,15,16]. Early attempts have already been made to determine the genetic determinants for the disease. Recent genome-wide association studies (GWAS) have identified several genes and single nucleotide polymorphisms (SNPs), associated with left displaced abomasum [2,17,18]. However, finding of new quantitative trait loci (QTL) and identification of the genes underlying this complex disease may lead both to a deeper understanding of LDA pathogenesis and the development of novel effective approaches to the treatment and prevention of this pathology.

Another effective tool for genomic analysis is the assessment of individual autozygosity of animals, or extended homozygous nucleotide fragments—runs of homozygosity (ROH), indicating inbreeding. The consolidation of ROH loci with a low frequency of recombination in generations of animals may indicate the formation of gene clusters under selection pressure, mostly due to positive selection for productivity traits, as well as adaptation to changing environmental conditions. ROH allow us to more effectively assess the degree of genomic inbreeding both in a population and individually, and also to evaluate the conservation of certain loci. A high degree of inbreeding in the studied animals can influence the results of genome-wide association studies, increasing the likelihood of detecting false positive associations. Moreover, the location of candidate genes in ROH may indicate their significant influence on the breeding quality of the animals. Modern industrial dairy farming technologies are based on the exclusive use of assisted reproductive technologies, such as artificial insemination of cows. For cryopreservation of genetic material (sperm), sires with the highest breeding value are selected. This approach leads to an increase in the inbreeding degree in herds and an increase in the population genetic homogeneity. As a result of the use of such a strategy for improving the genetic potential of dairy cattle, the frequency of predisposition hereditary factors for multifactorial diseases increases. Loci containing disease susceptibility factors often become homozygous, which increases the number of disease cases in herds.

The aim of this study is to find genetic associations characterizing the genomic variability of the predisposition to LDA in Holstein cattle of the Leningrad region of the Russian Federation.

## 2. Materials and Methods

### 2.1. Sampling and DNA Extraction

The collection of biological material for this study was carried out on three farms in the Leningrad region. This study analyzed two groups of animals, divided according to the principle of “case” and “control” (healthy animals). Peripheral blood samples were taken from a mixed-age herd of 360 highly productive Holstein cows in 2022, of which 175 animals belonged to the control group (animals with no history of LDA), and 185 cows belonged to the case group that consisted of animals with left displaced abomasum diagnosed by clinical examination. The diagnosis of LDA was made based on the results of percussion and auscultation in the area of the abomasum localization, and in some cases, laparotomy was used. The case group included all cases of left displaced abomasum that occurred during the study period.

Sampling was carried out in specialized vacutainer blood collection tubes with EDTA as an anticoagulant, which were then stored in a frozen condition at −20 °C until the DNA extraction procedure. DNA was isolated from samples using the phenol-chloroform method according to a standard proteolytic treatment protocol using proteinase K. Quality control of isolated DNA was carried out using a NanoDrop 2000 spectrophotometer (Thermo Fisher, Waltham, MA, USA).

### 2.2. Genotyping and Quality Control

For the genotyping procedure, the GeneSeek^®^ Genomic Profiler™ Bovine 100 K DNA chip (Neogen, Lansing, MI, USA) with a coverage density of 95,256 SNPs was used. Principal component analysis (PCA) was performed to assess population stratification and determine the number of genetic covariates. Quality control and filtering of genotyping data were carried out using the PLINK 1.90 software package [19]. When filtering the bioarray data, the following threshold values were used: the minimum allele frequency was set at a threshold of <0.05, the *p*-value threshold for the Hardy–Weinberg distribution was <1 × 10^−6^, the number of missing variants per sample was <0.05, and the proportion of missing variants per marker was also <0.05.

### 2.3. ROH Calling and Inbreeding Coefficient

The ROH analysis was performed using the detectRUNS package in R. Parameters for the ROH search were determined according to the following criteria [20]: a sliding window size of 25 SNPs, the proportion of homozygous overlapping windows was 0.05, a minimum of 35 homozygous SNPs included in a window, a maximum gap of 0.1 Mb between consecutive SNPs, and a maximum of one heterozygous SNP per window. The ROH islands were divided into three classes based on size: short (0.5–1 Mb), medium (1–4 Mb), and long (>4 Mb) [20]. To determine the inbreeding coefficient, we found the proportion of the genome covered by runs of homozygosity by dividing the total length of ROH by the length of the genome (FROH) [21] according to the following formula:(1)$${F_{ROH}}_{i}=\frac{\sum_{k=1}^{n}{L_{ROH}}_{i}}{L_{g}}$$
where:i represents the ith animaln stands for total number of ROH found in animal_i_,L_ROHi_ represents total ROH length for animal_i_,L_g_ identifies genome length of the individual.

We defined a common ROH island based on overlapping homozygous regions with a frequency greater than 0.50 for all animals, and also considered the suggestive frequency threshold greater than 0.40 to include more candidate genes and regions. Candidate genes located within ROH islands were identified using cattle genome assembly ARS-UCD1.3 (https://www.ncbi.nlm.nih.gov/datasets/genome/GCA_002263795.3/, accessed on 3 March 2023). To find genomic loci that may contain genes whose polymorphisms are strongly associated with the development of LDA, we also looked at the distribution of ROH islands in the case and control groups separately. The statistical significance of the difference in the incidence of ROH islands in healthy animals and cows with displacement was assessed using the Pearson chi-squared test.

### 2.4. Genome-Wide Association Analysis and Visualization

To ensure the accuracy of the genome-wide association study, we used three independent statistical models: General Linear Model (GLM), Mixed Linear Model (MLM), and Fixed and Random Model Circulating Probability Unification (FarmCPU), implemented in the GAPIT Version 3 [22] and rMVP [23] packages in R. GLM was conducted using population structure as a covariate, and in MLM, a relationship matrix was used as covariate. The fixed-effect model (FEM) in FarmCPU included the five significant principal components and pseudo-quantitative trait nucleotides. Genome-wide statistical significance threshold for all methods was determined using the Bonferroni correction [24] (α = 0.05).

The results were visualized using the qqman (v.0.1.9) [25] and qqplot2 (v.3.4.4.) packages in the RStudio programming environment (v. 2023.9.1.494) using the Manhattan plot.

### 2.5. Candidate Genes Identification

Annotation of the identified associations was carried out using cattle genome assembly ARS-UCD1.3, visualized in the genome browser GDV (Genome Data Viewer). To search for QTL associated with economically relevant traits, the CattleQTLdb database was used. For functional annotation of genes located within ±0.2 Mb of suggestive SNPs, the National Center for Biotechnology Information (NCBI) Gene database was used, as well as the open UniProt protein sequence database and scientific literature. The impact of a substitution on a protein sequence was determined using the VEP (Variant Effect Predictor) (v. 99) software [26].

## 3. Results

### 3.1. Genotype Filtering and Quality Control

Three-hundred-twenty-five samples passed quality control, and 85,605 polymorphic genetic variants were identified by genotyping with the GeneSeek^®^ Genomic Profiler™ Bovine 100 K (Illumina Inc, San Diego, CA, USA) DNA array. Of the filtered samples, 172 were from animals of the experimental group and 153 were from the control group.

### 3.2. Inbreeding Coefficient and Selection Signatures

We evaluated the inbreeding coefficient using the F_ROH_ method. The F_ROH_ ranged from 0.01 to 0.19 for individual genotypes and from 0.09 to 0.17 for mean F_ROH_ value for each bovine chromosome. We found that the distribution of homozygosity has similar values across autosomes. However, BTA9 and BTA15 are distinguished by a noticeably lower content of runs of homozygosity compared to other chromosomes. In contrast, BTA20 in animals from our sample on average contains more homozygous regions, which requires further consideration. The distribution patterns of genomic inbreeding coefficients among the studied animals and bovine chromosomes are shown in Figure 1.

The analysis of ROH island frequency in the dairy cattle population revealed several conserved genome regions, which we believe are under selection pressure. As a next step, we calculated the frequency of ROH and identified ROH islands for each chromosome in our population. First, we identified regions with an occurrence frequency greater than 0.5. Two such ROH islands were discovered on BTA14 and two were located on BTX. Information about the ROH islands is represented in Table 1. Only the most promising candidate genes were included in the corresponding column.

In addition, 20 homozygous regions were found, including at least 5 SNPs, which were found in more than 40% and less than 50% of the population under study. Suggestive ROH were mapped to BTA5, BTA7, BTA13, and BTA14, as well as BTA20, BTA21, BTA22, BTA26, and BTX. The longest suggestive homozygous regions were found on BTX (6.4 Mb, 5.5 Mb and 3.9 Mb). BTX also contained the largest number of such regions (*n* = three), which can be explained by the characteristics of recombination in sex chromosomes.

The distribution of ROH islands in the control and case groups of animals was different. In particular, four ROH, two of which were located on BTA13, one on BTA21, and one on BTX were more frequently represented in the genomes of animals in the control group. Among the ROH described above, the one located on BTA13 was significantly more common in animals with LDA. It is worth noting that no selection signatures were found that were significantly more common in the control group than in the experimental group. Information about the size of the loci, their frequency, and most promising candidate genes located in the ROH for both groups is shown in Table 2.

According to the CattleQTLdb, the first locus on BTA13 contains QTL associated with body conformation traits such as stature, dairy form, rear leg placement, body depth, rump width, foot angle, and traits related to productivity and susceptibility, namely, milk yield and milk fat yield and *M. paratuberculosis* susceptibility. The second ROH on the given chromosome, which has the largest difference in frequency between the two groups, also contains QTL for a similar set of traits, specifically body depth, rump width, stature, dairy form, udder cleft, and several others. Moreover, the occurrence of this region in the genomes of animals of the two groups was significantly different, which indicates the role of body shape in the development of the disease.

The region of homozygosity on BTA21 does not contain the described QTL, but in humans, loss of the homologous genomic region is connected with Prader–Willi syndrome. The ROH island on BTX is associated with semen quality, more specifically, percentage of normal sperm and general reproductive traits such as age at puberty and scrotal circumference.

### 3.3. Genome-Wide Association Study for Left Displaced Abomasum

Regarding the GWAS analysis, the choice of three models was based on their unique advantages in addressing various complexities of genetic analysis. GLM is an effective tool for analyzing simple population structures, whereas MLM provides a more robust framework for dealing with complex population structures, potentially reducing false positive associations [27]. FarmCPU takes a novel approach, implementing both fixed and random effects, allowing effects of cryptic relatedness and population stratification to be handled simultaneously, thus making it possible to map quantitative trait loci more accurately [28]. Our comparative analysis revealed subtle differences in the results of all three models. Thus, we focused on the MLM and FarmCPU models, as this could minimize the chance of finding false associations while increasing the likelihood of detecting true associations, resulting in a more robust and complete set of marker trait associations.

Figure 2 shows the results of a genome-wide association study with left displaced abomasum in the Holstein cows included in our study. At a nominal significance level of a −log_10_ *p*-value > 4.0, 14 SNPs were detected. The polymorphisms were identified on BTA2, BTA3, BTA6, and BTA9, as well as BTA16, BTA23, BTA24, BTA25, BTA27, and BTX.

The largest number of SNPs was located on BTA25 (*n* = 4); all polymorphisms were located within loci positioned from 17.7 to 27.1 Mb. Two nucleotide polymorphisms were located on BTA16, whereas the remaining chromosomes each contained one nucleotide substitution associated with the development of LDA.

### 3.4. Structural and Functional Annotation of Putative Genes for Left Displaced Abomasum

The structural annotation revealed 14 genes that contained detected SNPs or were close to them. Functional annotation of polymorphisms showed the presence of substitutions mainly in the non-coding parts of the genome (such as intergenic regions, introns). Of the 14 polymorphisms detected, 6 were in introns of the following genes: *ABCB11*, *SRP72*, *RGS18*, *GSG1L*, *FBXL19,* and *PRKCB*. The remaining polymorphisms were found in intergenic regions. Substitutions located in exons did not pass the threshold of statistical significance, were synonymous, and did not affect the functions of proteins encoded by the genes in which they were located. Further details about the associated polymorphisms with −log_10_ *p*-values greater than 4.0 can be found in Table 3.

Two SNPs were located in QTLs previously described for the trait. Thus, the SNP located on BTA23 was within the boundaries of the locus with coordinates 29.2–47.0 Mb [29]. Additionally, in previous studies on the Chinese population of Holstein cattle, one of the regions on chromosome 25 (24.8–26.8 Mb) was previously described for LDA [2].

GO enrichment analysis showed that candidate genes were involved in a wide range of biological processes, including protein ubiquitination (ABCB11, FBXL19), fatty acid metabolism (ABCB11, PNPLA4), the formation of organs and tissues as homeotic genes (SOX4), transcription regulation, regulation of signal transduction, and others. A list of GO (Gene Ontology) terms characterizing biological processes is displayed in the Table 4.

However, the biological role of certain genes has not been established due to their low representation in the scientific literature.

## 4. Discussion

Using strictly conservative methods for the significance threshold, such as the Bonferroni correction for all SNPs used in the present GWAS, there are no genome-wide significant results. A genome-wide association study for left displaced abomasum made it possible to identify associated polymorphisms localized close to or within genes, of which, genes such as *ABCB11*, *SRP72*, *RGS18*, *SOX4*, *GSG1L*, *FBXL19*, and *PNPLA4* are most promising. Our results suggest that susceptibility to LDA may be formed by the contribution of multiple genes with small effect sizes, consistent with moderate estimates of heritability for the trait.

Ricken et al. found genetic correlations between LDA and milk fat, milk protein, and milk yield [30]. Our results partly confirm previous findings, as some of the genes described in our study are located in QTL for traits such as milk fat content (*ABCB11*) [31] and glycerophosphocholine content (*GSGL1*) [32].

Runs of homozygosity analysis showed that the population we studied is characterized by a moderate degree of genomic inbreeding. For most animals, this value ranged from 0.1 to 0.15. Four ROH islands were discovered that were present in the genomes of most animals of both groups, all of them located on two chromosomes—BTA14 and BTX. Most likely, the reason for the appearance of such regions in the genome is selection pressure aimed at improving livestock productivity. It should be noted that some of the discovered regions, as well as the candidate genes located in them, have already been described for other herds in the scientific literature. One of the homozygous regions on BTA14 contains genes that have been associated with traits such as serum prolactin levels [33], growth and feed efficiency [34], carcass weight and eye muscle area in Korean Hanwoo cattle [35], and other carcass traits [36]. However, no production traits, particularly those important for dairy cattle, have been linked to this locus.

Some loci were more widely represented in the group of animals with LDA than in healthy individuals. Thus, the two loci on BTA13, both containing QTL for body conformation traits, had the greatest difference in prevalence (10.11% and 15.28%, respectively) between two groups. Moreover, the second ROH was statistically significantly different in its occurrence in the case and control groups, and was more widely represented in animals with LDA. That may be due to them containing selection signatures referring to the body conformation of Holstein cows. As previously mentioned, the occurrence of LDA was associated with an elliptical shape of the abdominal cavity section [9], which is typical for this breed of cattle. The shape of the abdominal cavity may affect the location of the internal organs, causing the abomasum to be pushed out of the right side of the body when it is inflated. Although body measurements were not taken in this study, we can hypothesize that the shape and volume of the abdominal cavity, according to genomic data, may be a predisposing factor for left displaced abomasum in Holstein cattle.

Another ROH more common in animals with LDA was located on BTA21. In our sample, healthy cows were 7.59% less likely to become its carriers. The locus contains five copies of genes coding small nucleolar *RNASNORD116* and one copy of *SNORD109A*. In humans, these genes are strongly associated with Prader–Willi Syndrome and Angelman Syndrome, complex multisystem genetic disorders with developmental delay, behavioral uniqueness, and speech impairment [37,38]. Prader–Willi Syndrome is also characterized by life-threatening obesity due to uncontrolled appetite and muscle weakness [38]. Despite the fact that QTL were not identified for production traits in this genetic region, the locus may play a significant role in the formation of the nervous system and appetite control both in humans and cattle according to the principle of orthology, which may be connected with the occurrence of LDA.

*ABCB11* encodes the bile acid export pump (BSEP), the main transporter of bile acids from hepatocytes to the biliary system [39]. The connection between liver diseases and impaired motility of the digestive system in cattle has been demonstrated in a number of publications [2,4,40,41]. Thus, in cows with pathologies of this organ, a decrease in the strength and duration of rumen contraction was noted [41]. Studies have reported that cases of LDA in animals with liver lesions are more common than in the general population [2]. When analyzing the biochemical profile of animals, it was found that in cases of LDA, markers of hepatocyte death significantly increase [4]. Vice versa, an increase in some hepatocyte death markers predicts the onset of displacement [11,12,42]. However, explanation of this connection requires further research. A previous GWAS study has already suggested a link between bile acid metabolism and LDA [2].

The *SRP72* gene encodes a polypeptide that is part of a protein complex that promotes post-translational modification of proteins in the endoplasmic reticulum [43]. In the signal recognition particle (SRP), SRP68 and SRP72 proteins form a heterodimer that has been difficult to investigate, and thus many processes in which the previously mentioned protein is involved still remain not fully understood. It is known that polymorphisms in *SRP72* are associated with one of the hereditary forms of aplastic anemia [44]. Transcriptomic studies have demonstrated that the expression of *SRP72* gene may be a marker of hepatotoxicity [45]. The mechanism of involvement of this protein in the development of left displacement of the abomasum is also not completely clear. SRP72 may carry out post-translational modifications of proteins, whose direct or indirect interaction with certain substrates will lead to a decrease in the force of smooth muscle contraction, which may affect peristalsis of the gastrointestinal tract and cause LDA. On the other hand, the effect may be more indirect. For example, it may be caused by changes in certain biochemical pathways.

*SOX4* encodes a transcription factor involved in the formation of several organ systems during embryogenesis [46]. The possible role of this gene in a hereditary predisposition to abomasum displacement is ambiguous, since *SOX4* is involved both in the formation of the nervous system and has a pleiotropic effect on the formation of the bile ducts of the liver [47].

The GSG1L protein is a subunit of the AMPA receptor that regulates the transmission of nerve impulses at synapses [48]. FBXL19 is involved in the ubiquitinylation of proteins and also regulates the activity of a number of homeotic genes by attaching CDK8 to their promoters [49], thereby participating in the formation of the musculoskeletal system. Internal organ disturbances in this process may be a predisposing factor for LDA. The protein is also involved in the IL-33 signaling pathway [50]. The role of these proteins in the development of left displaced abomasum requires further study.

Thus, we mapped loci that may be associated with LDA in Holstein cattle of the Leningrad region. The loci were identified on BTA2, BTA3, BTA6, and BTA9, as well as BTA16, BTA23, BTA24, BTA25, BTA27, and BTX. Most of the polymorphisms were not detected in previous studies, which may indicate the diversity of the genetic architecture in populations distributed in different geographical areas.

## 5. Conclusions

We focused on implementing a genome-wide association study and runs of homozygosity analysis to investigate the LDA trait in 360 highly productive Russian Holstein cows. Fourteen candidate SNPs were found to be nominally associated with left displacement of the abomasum at the threshold *p*-value of <1 × 10^−4^. ROH analysis revealed one ROH on BTA13 that was significantly more prevalent in the case group than in the control group of animals. This study revealed seven promising candidate genes for LDA susceptibility in the population under study: *ABCB11*, *SRP72*, *RGS18*, *GSG1L*, *FBXL19*, *PRKCB,* and *SPTLC*. Our findings may provide useful insights into the complex genetic architecture of predisposition to LDA in dairy cattle populations worldwide.

## Figures and Tables

**Figure 1 animals-14-02795-f001:**
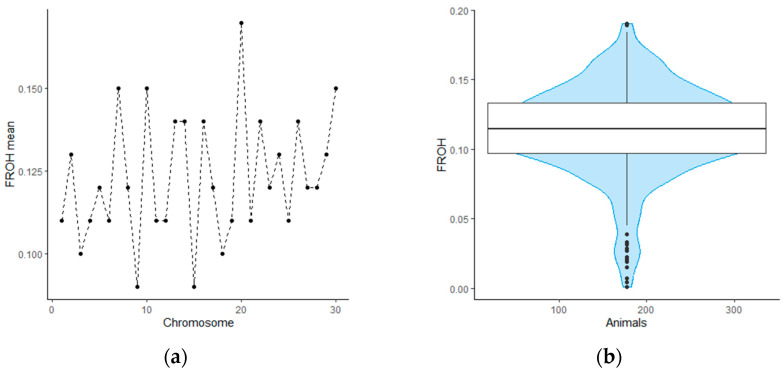
The distributions of inbreeding coefficient per individual and for each chromosome for highly productive Holstein cattle of Leningrad region: (**a**) line plot of mean inbreeding coefficient for each chromosome; (**b**) violin plot showing the distribution of the inbreeding coefficient in the studied population.

**Figure 2 animals-14-02795-f002:**
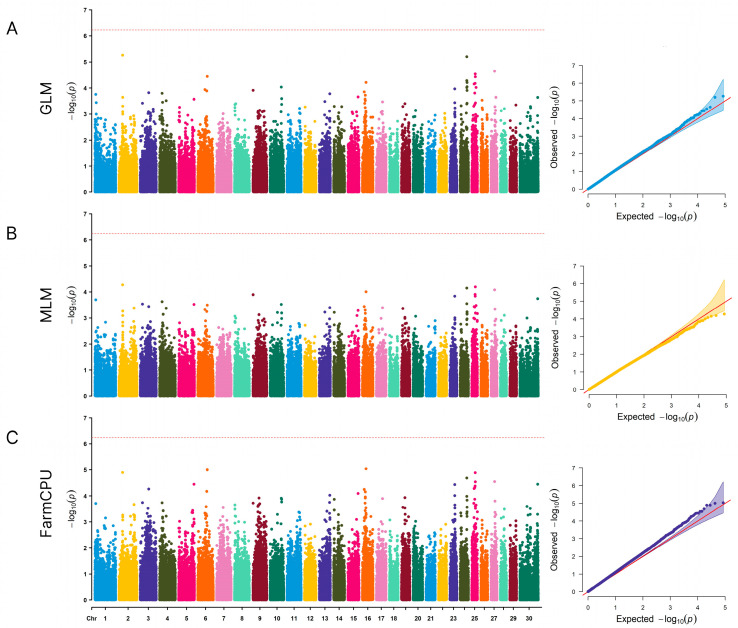
The Manhattan plots and quantile-quantile plots of GWAS results for left displaced abomasum in Russian Holstein cattle. (**A**) The Manhattan plots and quantile–quantile plots of the GWAS results for GLM. (**B**) The Manhattan plots and quantile–quantile plots of the GWAS results for MLM. (**C**) The Manhattan plots and quantile–quantile plots of the GWAS results for FarmCPU. The red horizontal line in the Manhattan plots represents genome-wide level of significance. The red diagonal line in the quantile–quantile plots represent expected −log_10_ *p*-values.

**Table 1 animals-14-02795-t001:** Genomic regions with a frequency of 50% or higher and key genes under selection pressure identified by ROH in Holstein cows.

Chromosome	Position, bp	Proportion of Animals with ROH in the Loci, %	Number of Genes	Candidate Genes
BTA14	22,750,647–23,430,999	89.92	9	*XKR4*, *TGS1*, *CHCHD7*, *PLAG1*
BTA14	23,542,871–23,630,896	53.22	1	*PENK*
BTX	65,046,412–69,995,413	70.86	25	*CYLC1*, *SATL1*, *POU3F4*, *POF1B*, *DACH2*
BTX	81,694,070–85,309,793	54.90	15	*AR*, *OPHN1*, *EDA2R*

**Table 2 animals-14-02795-t002:** Genomic regions and the most promising genes under selection pressure identified by ROH that show different occurrence in animals of control and case groups.

Chromosome	Position, bp	Frequency in Case Group Animals, %	Frequency in Control Group Animals, %	*p*-Value	Number of Genes	Candidate Genes
BTA13	5,228,105–5,988,846	40.11	30	0.11	5	*BTBD3*, *SNORA70*, *NOS2*
BTA13	6,289,502–6,580,508	41.75	26.47	0.03 *	1	*SPTLC3*
BTA21	1,988,047–2,044,107	50.54	42.94	0.35	6	*SNORD116*, *SNORD109A*
BTX	40,129,544–42,860,345	43.95	37.64	0.44	6	*PCDH11X*, *NAP1L3*, *FAM133A*

* *p* < 0.05 compared to healthy cows using the Pearson chi-squared test.

**Table 3 animals-14-02795-t003:** Significant SNPs associated by GLM with left displaced abomasum in Russian Holstein cattle.

SNP	MAF	BTA	Position	*p*-Value	Nearest Gene	Distance to Gene
BovineHD0200007887	0.47	2	27,152,746	2.6 × 10^−5^	*ABCB11*	Intron
BovineHD0300019207	0.46	3	64,155,288	8.5 × 10^−5^	*TRNAC-ACA*	+695,407
BovineHD4100004175	0.13	5	117,286,776	3.5 × 10^−5^	*MIR22850-5*	+52,582
BovineHD0600020458	0.07	6	71,848,355	1.8 × 10^−5^	*SRP72*	Intron
BovineHD1600003714	0.19	16	13,142,992	8.3 × 10^−5^	*RGS18*	Intron
ARS-BFGL-NGS-39696	0.37	16	26,927,766	1.4 × 10^−5^	*LOC132342448*	−2356
BovineHD2300010533	0.37	23	36,524,269	5.6 × 10^−5^	*SOX4*	+1731
BovineHD2400014124	0.38	24	50,082,597	3.9 × 10^−5^	*MAPK4*	−25,125
Hapmap40645-BTA-110440	0.42	25	25,480,228	2 × 10^−5^	*GSG1L*	Intron
ARS-BFGL-NGS-112441	0.37	25	27,035,533	4.3 × 10^−5^	*FBXL19*	Intron
BovineHD2500005062	0.34	25	17,812,656	8.2 × 10^−5^	*GP2*	−175,676
BovineHD2500006087	0.43	25	21,692,576	6.9 × 10^−5^	*PRKCB*	Intron
BTB-00965197	0.23	27	27,608,470	5.1 × 10^−5^	*PRRC2A-like*	+95,309
BovineHD3000042660	0.25	X	135,055,447	4.9 × 10^−5^	*PNPLA4*	+41,280

**Table 4 animals-14-02795-t004:** Genes containing associated at −log_10_ *p*-values > 4.0 SNPs or located close to them, and GO terms characterizing the biological processes in which they are involved.

Gene	GO ID	Term
*ABCB11*	GO:0015721	bile acid and bile salt transport
	GO:0015126	canalicular bile acid transmembrane transporter activity
	GO:0006631	fatty acid metabolic process
	GO:0120189	positive regulation of bile acid secretion
	GO:0016567	protein ubiquitination
*SRP72*	GO:0006614	SRP-dependent cotranslational protein targeting the membrane
*RGS18*	GO:0007186	G protein-coupled receptor signaling pathway
	GO:0009968	negative regulation of signal transduction
	GO:0008277	regulation of G protein-coupled receptor signaling pathway
*SOX4*	GO:0021782	glial cell development
	GO:0031018	endocrine pancreas development
	GO:0060485	mesenchyme development
	GO:0060563	neuroepithelial cell differentiation
	GO:0030182	neuron differentiation
*GSG1L*	GO:0099149	regulation of postsynaptic neurotransmitter receptor internalization
*FBXL19*	GO:0043161	proteasome-mediated ubiquitin-dependent protein catabolic process
*PNPLA4*	GO:0006357	regulation of transcription by RNA polymerase II
	GO:0055088	lipid homeostasis
	GO:0019433	triglyceride catabolic process

## Data Availability

The data presented in this study are available upon request.

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
