# Peer review of "Genome-Wide Association Study (GWAS) for Left Displaced Abomasum in Highly Productive Russian Holstein Cattle"

_animals, 2024, doi:10.3390/ani14192795_

Round 1
Reviewer 1 Report
Comments and Suggestions for Authors
1. The background section accounts for nearly half of the entire abstract. Is there too much introduction to the background section.
2. Should "-200C" in line 106 be "-200 ℃"? Check if some unit symbols (such as ℃, P, etc.) in the manuscript are written correctly, and if the same symbol is capitalized or italicized, the formatting should be consistent.
3. In lines 115-116, the minimum allele screening threshold is "<0.5". Should it be "<0.05".
4. Should chromosome 30 in Figure 1 be the X chromosome? Among the two threshold lines set, there are no significantly correlated SNPs on the stricter threshold line. Is it necessary to set this threshold line, and should the threshold line above the title in Figure 1 be p<1 × 10-6.
5. Can the Manhattan plots of the three methods in Figure 1 be displayed separately? It is recommended to add QQ plots for each method. Generalized mixed linear models may be more suitable for these data than general linear models and mixed linear models.
6. Should '[Huang H]' in line 258 be a reference but not labeled.
7. In lines 289-290, the abbreviations "carcass weight" and "eye muscle area" only appear once, and there is no need to label the abbreviations.
Author Response
- Comment 1.The background section accounts for nearly half of the entire abstract. Is there too much introduction to the background section.
Response 1: Thank you for the comment, we corrected the abstract (lines 18-21) - Comment 2. Should "-200C" in line 106 be "-200 ℃"? Check if some unit symbols (such as ℃, P, etc.) in the manuscript are written correctly, and if the same symbol is capitalized or italicized, the formatting should be consistent.
Response 2: Thank you for pointing this out. Line 109 in the manuscript has been revised. - Comment 3. In lines 115-116, the minimum allele screening threshold is "<0.5". Should it be "<0.05".
Response 3: Thank you for pointing this out. The minimum allele frequency filtering threshold was 0.05. We have revised line 121. - Comment 4. Should chromosome 30 in Figure 1 be the X chromosome? Among the two threshold lines set, there are no significantly correlated SNPs on the stricter threshold line. Is it necessary to set this threshold line, and should the threshold line above the title in Figure 1 be p<1 × 10-6.
Response 4: We have revised Figure 2. We changed the notation of the X chromosome and set the genome-wide significance level according to the Bonferroni correction. Also, we made the appropriate edits to the Materials and methods section (line 157-158). - Comment 5. Can the Manhattan plots of the three methods in Figure 1 be displayed separately? It is recommended to add QQ plots for each method. Generalized mixed linear models may be more suitable for these data than general linear models and mixed linear models.
Response 5: We have generated a novel Figure 2. We have added separate Manhattan plots for each statistical model and QQ plots for each method.
Thanks for your suggestion. We will definitely take it into account when conducting our next studies. In our current study we used several statistical methods for a multifaceted study of our data, according to the advantages of each model - Comment 6. Should '[Huang H]' in line 258 be a reference but not labeled.
Response 6: Thank you for pointing this out. Line 274 in the manuscript has been revised. - Comment 7. In lines 289-290, the abbreviations "carcass weight" and "eye muscle area" only appear once, and there is no need to label the abbreviations.
Response 7: We have revised lines 307-308.
Reviewer 2 Report
Comments and Suggestions for Authors
This article applied genome-wide association studies (GWAS) to the left displaced abomasum (LDA) disease in Holstein cattle, aiming to identify genetic associations that characterize the genomic variability related to the predisposition to LDA. In the method section, the study also conducted Runs of Homozygosity (ROH) analysis on two groups to assess the level of genomic inbreeding and the conservation of loci within the populations. The study ultimately succeeded in identifying candidate genes associated with the disease LDA, but there are a few comments need to be noted:
1. A nonnegligible problem with the manuscript is that it still contains many grammatical and word choice errors. The writing is poor enough in places to interfere with the understandability of the manuscript. Additionally, the logical relationship between some sentences is not clear. An example includes in line 71.
2. The structure of the manuscript needs to be improved. The description of “Method” and “Result” will be more clear by writing point-by-point.
3. The ROH and GWAS analysis were conducted in this study. However, the former was not mentioned in the summary and abstract.
4. The full name of an abbreviation must be specified the first time it appears.
5. It is more appreciated that if the amount of the data after QC was described at “Materials and Methods” section. The threshold of quality control for minor allele frequency(MAF) was <0.5 in this study. The size of MAF should range from 0 to 0.5, is it meat that the all SNP were retained or removed based on their MAF?
6. The materials and methods section lacks a description of the phenotypes used in the GWAS and the methodology for processing phenotypic data. Additionally, it is necessary to provide a detailed explanation of the GWAS model employed in the study.
7. In discussing the differences in ROH islands between the control and case groups, the article mentioned that four ROH regions were more frequently represented in the genomes of animals in the control group. However, the results in Table 2 contradict this conclusion.
8. Also in discussing the differences in ROH islands between the control and case groups, 4 ROH islands were list, but only one of them showed significant difference between case and control population. So why the remaining 3 ROH islands were list in the result or why the others were not mentioned in the result?
9. The serial number of the manhattan plot is “Figure 2”, not “Figure 1”.
10. The result of GLM in the manhattan plot should be removed if it was not be used. The separate plots of MLM and FarmCPU will be more suitable and clearer. Additionally, the Q-Q plots were needed to be provided.
11. The description of the FDR in the “Methods” section seems to be irrelevant to the following.
12. How is the threshold value (10-4) determined in the GWAS analysis? Ususally, it was set at 0.05/m or 0.01/m, and no significant associations would be detected in this threshold for this study.
13. The form of reference in line 258 needed to be correct.
Comments on the Quality of English LanguageThis article applied genome-wide association studies (GWAS) to the left displaced abomasum (LDA) disease in Holstein cattle, aiming to identify genetic associations that characterize the genomic variability related to the predisposition to LDA. In the method section, the study also conducted Runs of Homozygosity (ROH) analysis on two groups to assess the level of genomic inbreeding and the conservation of loci within the populations. The study ultimately succeeded in identifying candidate genes associated with the disease LDA, but there are a few comments need to be noted:
1. A nonnegligible problem with the manuscript is that it still contains many grammatical and word choice errors. The writing is poor enough in places to interfere with the understandability of the manuscript. Additionally, the logical relationship between some sentences is not clear. An example includes in line 71.
2. The structure of the manuscript needs to be improved. The description of “Method” and “Result” will be more clear by writing point-by-point.
3. The ROH and GWAS analysis were conducted in this study. However, the former was not mentioned in the summary and abstract.
4. The full name of an abbreviation must be specified the first time it appears.
5. It is more appreciated that if the amount of the data after QC was described at “Materials and Methods” section. The threshold of quality control for minor allele frequency(MAF) was <0.5 in this study. The size of MAF should range from 0 to 0.5, is it meat that the all SNP were retained or removed based on their MAF?
6. The materials and methods section lacks a description of the phenotypes used in the GWAS and the methodology for processing phenotypic data. Additionally, it is necessary to provide a detailed explanation of the GWAS model employed in the study.
7. In discussing the differences in ROH islands between the control and case groups, the article mentioned that four ROH regions were more frequently represented in the genomes of animals in the control group. However, the results in Table 2 contradict this conclusion.
8. Also in discussing the differences in ROH islands between the control and case groups, 4 ROH islands were list, but only one of them showed significant difference between case and control population. So why the remaining 3 ROH islands were list in the result or why the others were not mentioned in the result?
9. The serial number of the manhattan plot is “Figure 2”, not “Figure 1”.
10. The result of GLM in the manhattan plot should be removed if it was not be used. The separate plots of MLM and FarmCPU will be more suitable and clearer. Additionally, the Q-Q plots were needed to be provided.
11. The description of the FDR in the “Methods” section seems to be irrelevant to the following.
12. How is the threshold value (10-4) determined in the GWAS analysis? Ususally, it was set at 0.05/m or 0.01/m, and no significant associations would be detected in this threshold for this study.
13. The form of reference in line 258 needed to be correct.
Author Response
Comment 1: A nonnegligible problem with the manuscript is that it still contains many grammatical and word choice errors. The writing is poor enough in places to interfere with the understandability of the manuscript. Additionally, the logical relationship between some sentences is not clear. An example includes in line 71.
Response 1: Thank you for pointing this out. We have revised lines 69-72 as well as wording in lines 43-44, 46, 253 and 327.
Comment 2: The structure of the manuscript needs to be improved. The description of “Method” and “Result” will be more clear by writing point-by-point.
Response 2: Thanks for your suggestion. We added subheadings (lines 97, 114, 124, 150, 161, 171, 176, 230, 257)
Comment 3: The ROH and GWAS analysis were conducted in this study. However, the former was not mentioned in the summary and abstract.
Response 3: Thank you for pointing this out. Added have extended summary and abstract sections:
lines 14-15: «… In addition, we conducted runs of homozygosity analysis and…»
lines 24-26: «Runs of homozygosity analysis revealed one ROH on BTA13 that was found to be
significantly more prevalent in the group of animals with LDA than in the healthy group.»
Comment 4: The full name of an abbreviation must be specified the first time it appears.
Response 4: Thank you for the comment. We have revised lines 60, 63, 74-75
Comment 5: It is more appreciated that if the amount of the data after QC was described at “Materials and Methods” section. The threshold of quality control for minor allele frequency (MAF) was "0.5" in this study. The size of MAF should range from 0 to 0.5, is it meat that the all SNP were retained or removed based on their MAF?
Response 5: The threshold for MAF was "0.05", we have revised line 121.
Comment 6: The materials and methods section lacks a description of the phenotypes used in the GWAS and the methodology for processing phenotypic data. Additionally, it is necessary to provide a detailed explanation of the GWAS model employed in the study.
Response 6: Thank you, we have expanded the Materials and Methods section. Lines 101, 102, 104-107.
Comment 7: In discussing the differences in ROH islands between the control and case groups, the article mentioned that four ROH regions were more frequently represented in the genomes of animals in the control group. However, the results in Table 2 contradict this conclusion.
Response 7: Thanks for the comment. A tendency for greater prevalence of the four ROH in case group was revealed. For better presentation of the results, we highlighted the significantly associated ROH in Table 2.
Comment 8: Also in discussing the differences in ROH islands between the control and case groups, 4 ROH islands were list, but only one of them showed significant difference between case and control population. So why the remaining 3 ROH islands were list in the result or why the others were not mentioned in the result?
Response 8: We decided to describe all 4 ROH, since a tendency for their more frequent occurrence in case population was revealed, therefore we do not exclude their participation in the etiology of the pathology due to its complex inheritance.
Comment 9: The serial number of the manhattan plot is “Figure 2”, not “Figure 1”.
Response 9: We have revised line 248.
Comment 10:The result of GLM in the manhattan plot should be removed if it was not be used. The separate plots of MLM and FarmCPU will be more suitable and clearer. Additionally, the Q-Q plots were needed to be provided.
Response 10: Thank you very much for your comments. The results of the GLM are shown at Table 3 on page 7. We modified Manhattan plots and added Q-Q plots (Figure 2)
Comment 11: The description of the FDR in the “Methods” section seems to be irrelevant to the following.
Response 11: Thank you for the note, we have changed the approach to reducing probability of a Type I error by changing the FDR approach to a Bonferroni correction. This was reflected in the Materials and Methods section (lines 157-158), as well as on Figure 2.
Comment 12: How is the threshold value (10-4) determined in the GWAS analysis? Ususally, it was set at 0.05/m or 0.01/m, and no significant associations would be detected in this threshold for this study.
Response 12: Thank you for the comment. We set the genome-wide association threshold according to the Bonferroni correction. Given the multifactorial nature of the disease and, apparently, the complex nature of its inheritance we also set a lower threshold for putative associations at −log10p values < 4.0 level considering the Q-Q plots data in order to detect SNP with smaller effects on LDA.
Comment 13: The form of reference in line 258 needed to be correct.
Response 13: Thank you, corrected line 274.
Reviewer 3 Report
Comments and Suggestions for Authors
The document “Genome-Wide Association Study (GWAS) For Left Displaced Abomasum in Highly Productive Russian Holstein Cattle” presents an important topic in dairy cattle. Nevertheless, I strongly suggest adjusting the Bonferroni test to determine the threshold level, due to it is not a clear association in the association analysis.
For a better understanding of the studied population, it is important to explain how the animals (especially controls) were selected, and how many animals conformed to the total population.
In the analysis, I suggest performing GWAS using the ROH, and determining or confirming possible associations.
Figure 1 is difficult to appreciate and evaluate due to the chromosome length, but it could be included in a complementary graph to visualize the proportion of the chromosomes covered by ROH.
Tables 1 and 2. The authors include some genes and candidate genes, but the number and listed genes do not match. Include all genes or a note specifying that only the most important genes were included.
Line 197. I suggest presenting statistical results, and no subjective perceptions
Line 231, how do you define the significance threshold? I strongly recommend defining it with a statistical test.
I suggest reviewing the following paper:
o Talluri, R., Wang, J., & Shete, S. (2014). Calculation of exact p-values when SNPs are tested using multiple genetic models. BMC genetics, 15, 1-10.
Author Response
Comment 1:The document “Genome-Wide Association Study (GWAS) For Left Displaced Abomasum in Highly Productive Russian Holstein Cattle” presents an important topic in dairy cattle. Nevertheless, I strongly suggest adjusting the Bonferroni test to determine the threshold level, due to it is not a clear association in the association analysis.
Response 1: Thank you very much for your comment, to display results more accurately, we replaced the FDR with the Bonferroni correction and made appropriate adjustments in Materials and Methods section (lines 157-158) and also Figure 2.
Comment 2: For a better understanding of the studied population, it is important to explain how the animals (especially controls) were selected, and how many animals conformed to the total population.
Response 2: Thank you for pointing this out. We added relevant information about phenotypes to the Materials and Methods section (lines 101, 102, 104-107).
Information about the total population: farm #1 1800 cows, farm #2 2800 cows, farm #3 850
cows (without calves and heifers)
Comment 3: In the analysis, I suggest performing GWAS using the ROH, and determining or confirming possible associations.
Response 3: Thanks for reviewer’s suggestion, we`ll consider it for our next studies.
Comment 4: Figure 1 is difficult to appreciate and evaluate due to the chromosome length, but it could be included in a complementary graph to visualize the proportion of the chromosomes covered by ROH.
Response 4: Thanks for the comment. Due to the large number of animals and the number of chromosomes in the genome, plots containing detailed distribution of ROH by chromosomes for two groups of animals were not included in the article, but can be obtained as a separate file upon request, if needed.
Comment 5: Tables 1 and 2. The authors include some genes and candidate genes, but the number and listed genes do not match. Include all genes or a note specifying that only the most important genes were included.
Response 5: We agree with your comment, added clarifications (lines 194-197, 211-213). Only the most important genes were included.
Comment 6: Line 197. I suggest presenting statistical results, and no subjective perceptions
Response 6: Thank you, we corrected the wording (line 205-206)
Comment 7: Line 231, how do you define the significance threshold? I strongly recommend defining it with a statistical test.
Response 7: In order to explore the location of SNP with smaller effects on LDA we lower significance
threshold to level of −log10 p-values<4.0. When making this decision, we relied on the data from
the quantile-quantile plots, as candidate SNP show observed −log10 p-values above the expected
−log10 p-values according to GLM and FarmCPU.
However, to present the results more accurately, we have clarified the meaning of term
significance by changing wording in lines 243-244, 266-267, 281 and extending Discussion
section (lines 286-288)
Reviewer 4 Report
Comments and Suggestions for Authors
The authors prepared a manuscript describing the results of the conducted research very carefully. In the text itself, which is basically ready for printing, I noticed only minor editorial errors that should be corrected before printing. The authors' great professionalism is visible in the scope of conducting (complicated) genomic research, as well as describing the obtained results in the form of a publication. The work was quite easy to read, there is a sensible sequence of content, which allowed for a fairly precise reference to the GWAS results. I suggest publishing the work after making minor corrections, which I suggested directly in the pdf file.
Regards

Author Response
Thank you very much for your attention to our article and your positive comment on our work. We have made changes to the text of the article according to the comments.
Comment 1: was?
Response 1: Thank you for pointing that out. We have revised line 23: "...were.."
Comment 2: Genes` symbols in italic
Response 2: Thank you for your constructive suggestions. We have changed the formatting of the gene names in the lines 29-30, 261-262, 275-277, 289, 296, 324, 333, 344, 349-350, 358, 360 and Tables 1, 2 and 4
Comment 3: Please put in a proper citation for Nellore
Response 3: We have revised line 307.
Round 2
Reviewer 1 Report
Comments and Suggestions for Authors
This manuscript is acceptable.
Author Response
Thank you very much for your time and attention.
Reviewer 3 Report
Comments and Suggestions for Authors
The document improved significantly, especially in Material and methods, and improved organization.
Author Response
Thank you very much for your efforts to improve the manuscript.